# Obesity-Related Hypertension in Pediatrics, the Impact of American Academy of Pediatrics Guidelines

**DOI:** 10.3390/nu13082586

**Published:** 2021-07-28

**Authors:** Irene Rutigliano, Gianpaolo De Filippo, Luigi Pastore, Giovanni Messina, Carlo Agostoni, Angelo Campanozzi

**Affiliations:** 1Pediatrics, IRCCS Casa Sollievo della Sofferenza, San Giovanni Rotondo, 71013 Foggia, Italy; irene.rutigliano@libero.it; 2Department of Clinical and Experimental Medicine, University of Foggia, 71122 Foggia, Italy; giovanni.messina@unifg.it; 3Assistance Publique—Hôpitaux de Paris, Hôpital Robert-Debré, Service d’Endocrinologie et Diabétologie Pédiatrique, 75019 Paris, France; gianpaolo.defilippo@aphp.fr; 4French Clinical Research Group in Adolescent Medicine and Health, 75014 Paris, France; 5Pediatrics, Department of Medical and Surgical Sciences, University of Foggia, 71122 Foggia, Italy; luigiv.pastore@gmail.com (L.P.); angelo.campanozzi@unifg.it (A.C.); 6Department of Clinical Sciences and Community Health, University of Milan, 20122 Milan, Italy; 7Pediatric Intermediate Care Unit, Fondazione IRCCS Ca’ Granda Ospedale Maggiore Policlinico, 20122 Milan, Italy

**Keywords:** obesity, hypertension, children, cardiovascular risk

## Abstract

The prevalence of primary hypertension in pediatric patients is increasing, especially as a result of the increased prevalence of obesity in children. New diagnostic guidelines for blood pressure were published by the American Academy of Pediatrics (AAP) in 2017 to better define classes of hypertension in children. The aim of our study is to evaluate the impact of new guidelines on diagnosis of hypertension in pediatrics and their capacity to identify the presence of cardiovascular and metabolic risk. **Methods:** Retrospective clinical and laboratory data from 489 overweight and obese children and adolescents were reviewed. Children were classified according to the 2004 and 2017 AAP guidelines for systolic and diastolic blood pressure. Lipid profile and glucose metabolism data were recorded; triglyceride/HDL ratio (TG/HDL) was calculated as an index of endothelial dysfunction. Hepatic steatosis was detected using the ultrasonographic steatosis score. **Results:** Children with elevated blood pressure increased from 12.5% with the 2004 AAP to 23.1% with the 2017 AAP criteria (*p* < 0.001). There was a statistically significant increase in children with high blood pressure in all age groups according to the new cut-off values. Notably, the diagnosis of hypertension according to 2017 AAP criteria had a greater positive association with Hepatic Steatosis (rho 0.2, *p* < 0.001) and TG/HDL ratio (rho 0.125, *p* = 0.025). **Conclusions:** The 2017 AAP tables offer the opportunity to better identify overweight and obese children at risk for organ damage, allowing an earlier and more impactful prevention strategy to be designed.

## 1. Introduction

Over the last years, there has been increasing interest in childhood hypertension and greater recognition that many adult cardiovascular diseases have their origin in childhood. The childhood obesity epidemic had led to an increase in the prevalence of hypertension and its consequences in the young [1]. A recent examination of blood pressure (BP) and lipid levels in United States children clearly showed that the prevalence of elevated BP was grater in overweight and obese children that in the population as a whole [2]. Thus, it is evident that obesity represents one of the most important risk factor for development of primary hypertension even in pediatric population.

Pediatric hypertensive patients are usually asymptomatic, making this condition frequently misdiagnosed, but may present early manifestations of organ damage: 40% of patients among hypertensive children suffer from left ventricular hypertrophy and increased carotid intima-media thickness, an early marker of atherosclerosis [3]. Children with primary hypertension often remain hypertensive even in adulthood with possible irreversible consequences on health status: it has been reported that pediatric patients with high blood pressure have a risk of hypertension in adulthood increased by about 2.4 times [3]. In a large Italian cohort of 415 children and adolescents referred for obesity, 23.6% showed elevated blood pressure, according to criteria belonging to fourth report, the reference at the time of the study [4].

At the time, given the lack of outcome data, the current definition of hypertension in children and adolescents is based on the normative distribution of BP in healthy children. The lack of reliable data on the long-term consequences represents the major limitation of all attempts to define arterial hypertension in the pediatric age. Although it may seem intuitive that a non-physiological situation can lead to long-term damage and the more and more evidence is supported by the literature, it is important to design specific studies in pediatric populations to answer the question. The American Academy of Pediatrics (AAP) revised and published in 2017 new guidelines for the diagnosis of hypertension in children and adolescents, replacing the previous reference values published in 2004 [5,6]. New cut-offs have been proposed for the diagnosis in children <13 years while fixed thresholds independent of age, sex and height have been proposed for subjects ≥13 years of age. In particular, the revised guidelines have identified the diagnostic cut-off considering only the population of normal-weight children.

The aim of our study is to evaluate the impact of new guidelines on diagnosis of hypertension in pediatrics and their capacity to identify the presence of cardiovascular and metabolic risk.

## 2. Methods

### 2.1. Study Population

We retrospectively analyzed the data of 489 children and adolescents, all Caucasian, admitted to Pediatric Outpatient Clinic of University of Foggia (Italy) for overweight including obesity, in the period between February 2008 and December 2013. Overweight was defined as body mass index (BMI) ≥ 85° and <95° percentile and obesity as a BMI ≥ 95° percentile for age and sex, according to the growth curves of the Center for Disease Control [7].

The presence of secondary forms of excess weight or other chronic diseases has been considered a criterion for exclusion. Anamnestic information, auxological data (weight, height, waist circumference and pubertal stage) were collected for all subjects according to standard procedures [8].

Since BMI is gender- and age-dependent, the z-score BMI was calculated for all analyzed subjects, which is a measure completely independent from these variables. According to CDC 2000 standards, the LMS method was used (statistical method to normalize data distribution, where L = Box–Cox power, M = generalized mean of data and S = standardized variation coefficient) [9]. The waist circumference/height ratio (WtHr, waist to height ratio) was evaluated for each patient, being an index of visceral adiposity [4,10].

Systolic and diastolic blood pressure was obtained for each child. The blood pressure measurement was performed in a seated patient, after at least five minutes of rest, with differential cuffed aneroid sphygmomanometer (auscultatory method) according to the age and constitution of the child. Three blood pressure measurements were taken for each patient and the average value recorded for the analysis [6,11].

In a first time, absolute values were converted to age-, gender-, and height-specific percentiles, using tables provided by the 2004 National High Blood Pressure Education Program Working Group on High Blood Pressure in Children and Adolescents (2004 AAP) and based on these data, we calculated the blood pressure z-score [5]. The same data were then analyzed in accordance with the new 2017 AAP criteria [6].

For children with abnormal values of blood pressure, second-level clinical-laboratory and instrumental investigations were carried out to exclude secondary hypertension. Only data of children without evidence of secondary hypertension were included into the study. The study was approved by the Ethics Committee of the University Hospital of Foggia.

### 2.2. Laboratory Analysis

Venous blood samples from each subject were taken in the morning, after at least 10 h of fasting. Blood glucose was determined using the gluco-oxidase method.

Total cholesterol, HDL cholesterol, LDL cholesterol and triglycerides were determined using automated enzymatic methods (Unicel DxC800 SynchronTM System, Beckman Coolter, Fullerton, CA, USA). Insulin was measured by the ELISA method (Elecsys, Roches diagnostics, Mannheim, Germany). The HOMA index (Homeostasis Model Assessment) was calculated as an insulin resistance index [12].

The triglycerides/HDL ratio was also calculated as a marker of endothelial dysfunction [13,14].

### 2.3. Hepatic Ultrasound

All subjects underwent liver ultrasound examination after a minimum of 10 h of fasting, using a high-resolution ultrasound system (LOGIQ^®^ 7, GE Medical Systems, Milwaukee, WI, USA). The level of liver echogenicity was graded according to the ultrasonographic steatosis score [14]. Subjects were classified as “with and without hepatic steatosis” according to ultrasonographic findings [15,16].

### 2.4. Statistical Analysis

Results are expressed as mean ± standard deviation (SD) with 95% confidence interval for continuous variables, as percentages for categorical and discrete variables. For the non-Gaussian distribution parameters, instead, the median and the interquartile range were used. The Kolmogorov–Smirnov test was applied to test the hypothesis of normality of the data. The data were analyzed by t-Student test, Mann–Whitney U test, χ^2^ test. The calculation of Spearman’s rho coefficient was used to evaluate the degree of association between variables. Values of *p* < 0.05 were considered statistically significant.

## 3. Results

The study population consisted of 489 children, mean age 9.4 ± 2.5 years (range 3–15.8 years): 270 boys (mean age 9.4 ± 2.5 years, range 3–15.8 years) and 219 girls (mean age 9.3 ± 2.5 years, *p* = 0.668). Their mean BMI z-score was 2.3 ± 0.49 (range 1.64–5.62), in particular 2.4 ± 0.6 for boys and 2.2 ± 0.3 for girls (*p* = 0.001).

The median value of systolic pressure z-score in the total population was 0.33 (IQR −0.19–0.89): in male subjects 0.38 (IQR −1.45–1), in female subjects 0.35 (IQR −0.23–0.88, *p* = 0.257). The diastolic pressure z-score was 0.4 ± 0.79 in the total population: 0.4 ± 0.8 in males and 0.38 ± 0.78 in females (*p* = 0.531).

### BP Values Classification According to 2004 and 2017 AAP Criteria

According to the 2004 AAP criteria, 84.7% of values showed normal systolic blood pressure, 5.9% were in pre-hypertensive state and 9.4% were classified as systolic hypertensive state. Regarding the diastolic pressure: 87.7% of values were in normal range, 5.7% were in pre-hypertension status, 6.5% were in the range of diastolic hypertension. From the overall statistical analysis conducted: 87.5% of values were in the normal range, 6.1% suggested systolic hypertension, 3.3% diastolic hypertension, 3.1% systodiastolic hypertension according to 2004 AAP cut-offs.

According to the latest guidelines (2017 AAP): 71.2% values showed normal systolic blood pressure, 13.5% were in the range of higher systolic blood pressure, 12.5% of hypertensive Stage 1, 2.9% of hypertensive Stage 2. Regarding diastolic pressure: normal diastolic values were found in 78.9% 9.2% were in high diastolic range, 10.6% in Stage 1, and 1.2% in Stage 2. In particular: 76.9% systodiastolic values were in normal range 11% were in hypertensive range only for systolic, 7% only for diastolic and 5.1% for both.

Then there was a shift from 12.5% to 23.1% of total hypertensive values from the old to the new classification (*p* < 0.001), in particular from 6.1% to 11% for systolic blood pressure only, from 3.3 to 7% for diastolic blood pressure only, from 3.1 to 5.1% for systodiastolic hypertension range.

At the valuation with the new cut off: only 87.9% of the values classified as not hypertensive according to 2004 AAP remained so, while 52 values reclassified as hypertensive (equal to 12.1%, average age 10.1 ± 2.6 aa, found in 30 males and in 22 females): 28 values reclassified as systolic arterial hypertension, 20 values as diastolic hypertension and 4 as systodiastolic hypertensive state.

In particular, by analyzing the population according to quartiles by age, listed in Table 1, we found in the 2004 AAP classification the following subjects harboring values in hypertensive range: 16.5% in I, 11.4% in II, 10.6% in III, 11.5% in IV (*p* = 0.489). With the 2017 AAP classification, instead, 22.3% in I, 20.3% in II, 22.8% in III, 27% in IV (*p* = 0.647).

The subjects recognized has having high blood pressure according to AAP 2017 but classified has normotensive in AAP 2004 were 7/121 in the I quartile (5.8%), 11/123 in II (8.9%), 15/123 in III (12.2%), 19/122 in IV (15.6%). The increase in the percentage of subjects carrying abnormal values according to the quartile by age, was statistically significant at the trend test (*p* = 0.009). Of the 52 with abnormal values with new classification, 17.3% suffered from hepatic steatosis. Children with ultrasonographic signs of hepatic steatosis are listed in Table 2.

Values falling in the hypertensive range according to 2004 AAP showed positive and statistically significant correlation with WtHR (rho 0.147, *p* = 0.001), with liver steatosis (rho 0.131, *p* = 0.004), but not with HOMA index (*p* = 0.181) nor with TG/HDL ratio (*p* = 0.153).

Values falling in the hypertensive range according to 2017 AAP had positive and statistically significant correlation with WtHR (rho 0.122, *p* = 0.004), with HOMA index (rho 0.103, *p* = 0.022), with liver steatosis (rho 0.2, *p* < 0.001) and with TG/HDL ratio (rho 0.125, *p* = 0.025). Data on clinical and metabolic parameters of children divided according to AAP 2004 and AAP 2017 blood pressure values are expressed in Table 3.

## 4. Discussion

Hypertension is one of the main causes of mortality worldwide [17]. In the past it was considered a concern for adulthood only, but the current evidence has instead allowed us to understand that essential hypertension can also affect the pediatric age [18]. The atherosclerotic process, as now widely validated in the literature, can start early in children and high blood pressure levels are part of the etio-pathogenetic mechanism of vascular alterations which, as clearly known, are then responsible for the main causes of mortality and cardio-vascular morbidity in adulthood [19,20].

Paying attention to blood pressure in children and adolescents allows, in fact, to operate large-scale prevention of cardiovascular disease [21,22]. There are also several combinations of factors that play a pathogenetic role in the onset of essential hypertension and these risk factors are divided into “non-modifiable”, such as age, gender and familiarity, and “modifiable” factors, first of all obesity [23]. For this reason, AAP has decided to re-evaluate the diagnostic cut-offs for hypertensive status in pediatrics, considering only normal-weight patients for the construction of new nomograms [6]. Thus, the limits of the diagnosis of hypertension in pediatric age have been significantly reduced, with substantial and important changes.

Within the same population, as we demonstrated in our results, the percentage of values falling in range of hypertension increased significantly from 12.5% of the 2004 AAP criteria to 23.1% for the 2017 AAP criteria. The 2017 AAP tables, therefore, allowed us to classify more values as hypertensive than the previous assessment. In particular, the highest increase involved older children within our sample. This type of intervention improves, therefore, the recognition of children requiring closer clinical follow-up in order to make the necessary lifestyle changes and improve their health outcomes [24].

In fact, in our analysis the finding of hypertensive values according to the 2017 AAP criteria was more significantly associated with cardio-metabolic risk parameters such as liver steatosis (rho: 0.2 vs. 0.131) and, in particular, had a statistically significant association with HOMA index, an insulin resistance parameter, and with the TG/HDL ratio, an indirect endothelial dysfunction index within the same population of overweight or obese children [13,14]. This association was not detected using the 2004 classification.

Our data show that even in younger children, those of I quartile with age <7.75 years, we found a percentage of more than 20% of subjects harboring abnormal values, while the percentage was even close to 27% in IV quartile subjects. Since the most frequent form of hypertension in pediatrics is the secondary one, especially in younger children, it is suggestive to imagine that the increase of hypertensive values found in in our population is not only related to the increased sensitivity of the new diagnostic guidelines, but also to the spread of the epidemic of obesity with a consequent change in the epidemiology of hypertension in pediatric age [25]. The use of the new guidelines seems to offer the possibility to better screen the pediatric population, recognizing the subjects at higher cardiovascular risk and with signs of endothelial dysfunction [26]. A rather high percentage of overweight children had hypertensive status according to both 2004 AAP and 2017 AAP guidelines. A prevalence of hypertensive arterial hypertension in overweight children is reported from 4 to 23% (4–14% in overweight and 11–23% in obese subjects), mainly confirmed by our results.

The strengths of the study were to investigate the correlation of BP values with objective markers of metabolic or vascular dysfunction (i.e., liver steatosis, HOMA index, TG/HDL ratio), not limiting the study to a retrospective observation according to new criteria. Furthermore, the population was ethnical homogenous, allowing to avoid any confusion factor belonging to different prevalence in non-European populations.

The major limit is the single evaluation of BP. In effect, a correct diagnosis of BP state (i.e., normal, pre-hypertensive and hypertensive) need at least three consecutive evaluations to classify the subject [6]. The retrospective design does not allow such analysis. The reasoning therefore revolves around the concept of abnormal value of blood pressure and not of hypertensive subject, for methodological correctness. In fact, this specificity that can at first be interpreted as a limit in the study (the fact of not being able to classify subjects as normal and hypertensive in terms of clinical diagnosis) can also be understood as a force of the study. In fact, the correlation with the markers of metabolic and endothelial risk is already evident in subjects with only one pathological measurement (and that therefore will not necessarily be confirmed as hypertensive to a subsequent deepening). It is, therefore, legitimate to think that the correlation between pathological blood pressure values and organ damage can be even stronger by restricting the analysis to patients with consistently high values. This is the purpose of future research. Moreover, our analysis was set on obese and overweight children so we cannot say that the results can be extrapolated to the general pediatric population, including normal weight subjects. In conclusion, the 2017 AAP tables offer the possibility to identify earlier children at risk for organ damage, allowing the structuration of more incisive prevention strategy.

## Figures and Tables

**Table 1 nutrients-13-02586-t001:** Percentile according to age in our population.

Percentile	Age (yrs)
25	7.7
50	9.4
75	11

**Table 2 nutrients-13-02586-t002:** Children with ultrasonographic signs of hepatic steatosis according to sex.

Sex	No Hepatic Steatosis	Hepatic Steatosis	Total
Males n (%)	247 (91.5)	23 (8.5)	270
Females n (%)	206 (94.1)	13 (5.9)	219
Total n (%)	453 (92.6)	36 (7.4)	489

**Table 3 nutrients-13-02586-t003:** Median value (inter quartile range) and subjects with liver steatosis in children classified according to AAP 2004 and AAP 2017 blood pressure values.

Classification	AAP 2004 Hypertension		AAP 2017 Hypertension	
	No	Yes	No	Yes
WHtR	0.56 (0.53–0.6)	0.59 (0.55–0.65)	0.56 (0.53–0.6)	0.57 (0.54–0.64)
TG/HDL	1.6 (1.1–2.5)	1.8 (1.2–2.5)	1.6 (1.1–2.4)	1.8 (1.2–2.7)
HOMA	2.7 (1.9–3.8)	3 (1.9–4.3)	2.7 (1.9–3.7)	3.2 (1.9–4.3)
Liver Steatosis %	6.1	16.4	4.5	16.8

## Data Availability

Data are available on demand.

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
