# Peer review of "Obesity-Related Hypertension in Pediatrics, the Impact of American Academy of Pediatrics Guidelines"

_nutrients, 2021, doi:10.3390/nu13082586_

Round 1
Reviewer 1 Report
The aim of the study is to evaluate the impact of new guidelines (AAP 2017) on diagnosis of hypertension in pediatrics and their capacity to identify the presence of cardiovascular and metabolic risk. The authors retrospectively analyzed the data of 489 children and adolescents, all Caucasian, admitted to Pediatric outpatient clinic of University of Foggia (Italy) for overweight including obesity. The conclusion was that the 2017 AAP tables offer the opportunity to better identify overweight and obese children at risk for organ damage, allowing an earlier and more impactful prevention strategy to be designed.
Even though the study is focused on an interesting topic there are major concerns:
Repeated measurement of BP remains an important step in determining whether a child truly has HBP. A single BP measurement may lead to a misclassification of BP category and inappropriate action. Repeated BP measurement remains a requirement for diagnosing HTN.
In those subjects that were classified as having high BP according to AAP 2017 but normotensive according to the 4th Task Force, what are the results regarding the classical TOD assessment? Previous studies have demonstrated that in obese children and adolescents the application of more 'physiological' nomograms, based on a population of normal-weight children, did not yield any advantage in identifying individuals with early cardiac organ damage. Antolini L et al. J Hypertension 2019;37:1213-1222. How many of these subjects required antihypertensive drug treatment?
Concerning the US, despite its widespread use, liver US has several important limitations that healthcare providers should recognize, particularly its low sensitivity. The methodology applied for the diagnosis of hepatic steatosis needs to be discussed more in depth. Recent bibliography about techniques and diagnosis of NAFLD has been published.
Author Response
We thank Reviewer 1 for the opportunity to improve our manuscript with his comments and suggestions. Below, our point-by-point answers (in red).
Repeated measurement of BP remains an important step in determining whether a child truly has HBP. A single BP measurement may lead to a misclassification of BP category and inappropriate action. Repeated BP measurement remains a requirement for diagnosing HTN.
Measurements were taken several times in each individual subject on the same day, after a rest period of at least 5 minutes. An average was taken. It is a retrospective study and it is not intended to evaluate different groups of subjects, but in the same subject under the same conditions we evaluated what changes in terms of blood pressure with the old and the new AAP system. We didn't even make the diagnosis of hypertension but simply classified and reclassified the subjects according to old and less old cut offs, so we could see what changed in the definition of cardio metabolic risk characteristics.
In those subjects that were classified as having high BP according to AAP 2017 but normotensive according to the 4th Task Force, what are the results regarding the classical TOD assessment?
Of the reclassified subjects, 17.3% had hepatic steatosis, 9.9% had hypertransaminasemia and 30.8% had a cardiometabolic score ≥ 2, while 36.5% had a TG/HDL ratio ≥ 2.
Previous studies have demonstrated that in obese children and adolescents the application of more 'physiological' nomograms, based on a population of normal-weight children, did not yield any advantage in identifying individuals with early cardiac organ damage. Antolini L et al. J Hypertension 2019;37:1213-1222. How many of these subjects required antihypertensive drug treatment? The study is a retrospective one; there is no intention to diagnose high blood pressure but only to understand any advantages (or disadvantages) of using more restrictive normograms in obese subjects.
Concerning the US, despite its widespread use, liver US has several important limitations that healthcare providers should recognize, particularly its low sensitivity. The methodology applied for the diagnosis of hepatic steatosis needs to be discussed more in depth. Recent bibliography about techniques and diagnosis of NAFLD has been published.
Certainly the ultrasonographic technique is not the gold standard for the diagnosis of NAFLD. Biopsy would be the gold standard, but it is far from being used in the context of our study, which aims to compare old and new guidelines. Certainly the US examination is operator dependent, but in our case the use of a single operator for the analysis of children eliminates inter-individual variability in the first place. It should also be noted that for an experienced operator, US examination shows steatosis when liver fat levels are > 30%, i.e. when steatosis is well present. Thus, US examination is likely to underdiagnose rather than overdiagnose HS. In our case, the US examination nevertheless made it possible to recognise subjects with indirect signs of significant liver damage and, in some cases, associated with uncertain transaminase levels (9.9% of subjects reclassified with elevated levels). In addition, 19.4 per cent of subjects with diagnosed steatosis in the US had hypertransaminasemia.
Reviewer 2 Report
Dear Authors,
This paper concerned problem of increasing obesity and related with it problems of hypertension in pediatric population. This is interesting and important topic. But I want to underline a few problems which I saw:
- In my opinion title of manuscript is not very proper. Authors analysed only group of overweight and obese children in the study is no information about whole paediatric population, from this could not conclude about all population. Authors did not demonstrated increasing hypertension dependently on increasing BMI from normal or lean to pathological obesity. I suggest changing the title, highlighting the strengths of the study- comparing the ranges in two guidelines or maybe pointing the identification of risk factors. For example: The impact of changing AAP guidelines on frequency of hypertension diagnosis in overweight and obese children.
- Were all analysed parameters not normally distributed? - Authors use only Spearman R Rank correlation which is poorer in strength than Pearson correlation matrices?
- Why You use shortcut ”rho”- the most frequently is used “r” coefficient.
- In the manuscript authors highlights statistically significant correlations, but these results are very weak. The coefficient r<0,2 rather pointed on no dependence (especially in Spearman Rank correlation), and r=0,2-0,4 on weak dependence. Please check it with Your statistician and take back to these results once more in in the manuscript.
Author Response
We thank Reviewer 2 for the opportunity to improve our manuscript with his comments and suggestions. Below, our point-by-point answers (in red).
This paper concerned problem of increasing obesity and related with it problems of hypertension in pediatric population. This is interesting and important topic. But I want to underline a few problems which I saw:
In my opinion title of manuscript is not very proper. Authors analysed only group of overweight and obese children in the study is no information about whole paediatric population, from this could not conclude about all population. Authors did not demonstrated increasing hypertension dependently on increasing BMI from normal or lean to pathological obesity. I suggest changing the title, highlighting the strengths of the study- comparing the ranges in two guidelines or maybe pointing the identification of risk factors. For example: The impact of changing AAP guidelines on frequency of hypertension diagnosis in overweight and obese children.
The aim of the study is not to compare obese and non-obese children, but simply to assess what changes occur in the same group of subjects using old and new pressure classification systems. What benefit is achievable? What’s the advantage? Each subject becomes his or her own comparison: from the AAP 2004 to the AAP 2017 analysis. Do the new cut-offs allow us paediatricians to recognise earlier subjects at greater risk, and therefore subjects who deserve a more intensive approach and follow-up?
Were all analysed parameters not normally distributed? -
We evaluated the distribution of the data with the kolmogorov smirnov test. Gaussian data are all expressed as mean and standard deviation. Non-Gaussian data, in median and IQR. For the correlation, the analysed data are WtHr, HOMA index, hepatic steatosis, TG/HDL ratio: they are all non-Gaussian (steatosis yes and no is also a categorical variable) so the suitable bivariate correlation test is the Spearmann test. We also tried to transform the quantitative data into Gaussians (with logarithmic transformation) but they remain non-parametric.
The statistical analysis was conducted between the above parameters and the diagnosis of high blood pressure (yes or no) based on AAP 2004 and AAP 2017 (not the absolute value), therefore the latter two parameters (hypertension yes or no) are not quantitative but qualitative (or categorical) variables and therefore can be analysed with non-parametric tests such as the Spearman test. Pearson's test is a parametric test.
For correlation analysis between steatosis and hypertension diagnoses according to AAP 2004 and 2017 we also confirmed the correlation figure with Pearson's chi square and likelihood ratio: for AAP 2004 likelihood ratio 6.649 with p=0.01, for AAP 2017 likelihood ratio 16.3 with p<0.001.
Authors use only Spearman R Rank correlation which is poorer in strength than Pearson correlation matrices?Why You use shortcut ”rho”- the most frequently is used “r” coefficient.
The spearman test is a bivariate correlation, the coefficient is called rho to recognise it from Pearson's r, alternatively someone uses R with small s at the base (even the SPSS otput reports Speramn's rho)
In the manuscript authors highlights statistically significant correlations, but these results are very weak. The coefficient r<0,2 rather pointed on no dependence (especially in Spearman Rank correlation), and r=0,2-0,4 on weak dependence. Please check it with Your statistician and take back to these results once more in in the manuscript.
The results of the correlation analysis are reported purely descriptively, without any causal relationship. An increase in the number of high blood pressure diagnoses in the new classification could allow early recognition of children deserving assessment and close follow-up.
What is essentially highlighted in the discussion is the fact that only with the diagnosis according to AAP 2017 there is the value of significance in the analysis of association with hepatic steatosis that reaches an index of 0.2 (p<0.01), a relationship also confirmed by the Pearson chi-square. In the other cases, moreover, although the association index is weak, it indicates a direction of positivity of the relationship with significance in the case of HOMA index and WtHR (rho is low, but the direction of the relationship is positive and significant) only for AAP 2017 diagnoses.
Round 2
Reviewer 1 Report
The authors appropriately answered the inquiries made by the reviewer.